

# Estimation of surface depression storage capacity from surface roughness

Mohamed A. M. Abd Elbasit[1,2], Chandra S. P. Ojha[3], Majed M Abu-Zerig[*4,5], Hiroshi Yasuda[4], Liu Gang[6], Fethi Ahmed[2]

[1]Agricultural Research Council- Soil Climate and Water, Private Bag X79, Pretoria 0001, South Africa.
[2] *School of Geography, Archaeology, and Environmental Studies, University of the Witwatersrand, Johannesburg 2000, South Africa*
[3] Dept. of Civil Engineering, Indian Institute of Technology, Roorkee, India
[4]International Platform for Dryland Research and Education, Tottori University, Tottori, Japan. drmajedo@gmail.com
[5]Civil Engineering Department, Jordan University of Science and Technology, Irbid, Jordan. Email: majed@just.edu.jo
[6]*State Key Laboratory of Soil Erosion and Dryland Farming on the Loess Plateau, Institute of Soil and Water Conservation, Northwest A&F University, Yangling 712100, People's Republic of China*

*Correspondence to:* Majed M Abu-Zreig (majed@just.edu.jo; drmajedo@gmail.com)

**Abstract.** Depression storage models found in the literature were developed using statistical regression for relatively large soil
surface roughness and slope values resulting in several fitting parameters. In this research, we developed and tested a conceptual model to estimate surface depression storage having small roughness values usually encountered in rainwater harvesting microcatchments in arid regions with only one fitting parameter. Laboratory impermeable surfaces of 30 x 30 cm2 were constructed with four sizes of gravel and mortar resulting in random roughness values ranged from 0.9 to 6.3 mm. A series of laboratory experiments were conducted under 9 slope values using simulated rain. Depression storage for each
combination of relative roughness and slope were estimated by mass balance approach. Analysis of experimental results indicated that the developed linear model between DSC and the square root of the ration of random roughness (RR) to slope was significant at probability value of 0.001 and coefficient of determination $R^2$=0.90. The developed model predicted depression storage of small relief at higher accuracy compared to other models found in the literature.

**Keywords:** Impermeable surface; micro-flow simulator; surface roughness; arid region.

# 1 Introduction

Surface depressions and micro-relief play a significant role in surface runoff and sediment yield of agricultural fields (Dunne et al., 1991; Hairsine et al., 1992; Huang and Bradford, 1992; Govers et al., 2000; Takken et al., 2001; Darboux et al., 2002). However, quantifying this storage on the watershed and field scale is one of the most challenging tasks for scientists. Therefore, estimating the depression storage capacity (DSC) of land has continued to receive wide attention from researchers worldwide.



The surface storage can be classified into depression storage and surface detention (Stammers & Ayers, 1957). The difference between the depression and detention storages is that the depression storage does not contribute in the recession area in the hydrograph. On the other hand, the detention storage has two possibilities: to infiltrate or/and to appear as recession part of the hydrograph (Fig. 1). Mathematically, this concept can be explained by the water mass balance. Geometrically, soil depression is subsidence/puddle points on soil surface compared to adjacent/surrounding points. The depressions storage is a depth of water in soil pits (Antoine et al., 2012). Soil depression storage capacity (DSC) is the maximum depth of water that can be stored in soil depression areas.

Depression storage capacity is affected by wide range of parameters related to soil, topography, agricultural practices and environmental conditions. To simplify such complex set of factors, researchers have been trying to develop soil roughness indices that best describe or quantify DSC for land surfaces. Soil roughness is a measure of the variations in surface elevation. The research has intensified by the development of digital and laser scanning techniques that enable researchers to obtain digital elevation map thus estimating soil surface roughness parameters with proper computer algorithm in relatively precise and short time. Therefore, calculating DSC based on well-estimated soil roughness parameters become possible (Mitchell and Jones, 1976, 1978; Moore and Larson, 1979, Ullah and Dickinson, 1979; Kamphorst and Duval, 2001; Abedini, 1998; Abd Elbasit et al., 2009).

Several soil roughness parameters have been developed by researchers to predict DSC accurately for a given soil condition (Onstad, 1984; Linden et al., 1986; Zobeck and Onstad, 1987; Sneddon and Chapman, 1989; Huang and Bradford, 1990; Mwendera and Feyen,1992; Hansen et al., 1999; Kamphrost et al, 2000; Borselli and Torri, 2010). Those parameters include, random roughness (RR), tortuosity, limiting elevation difference and slope and the mean upslope depression. The relative roughness (RR) soil surface seemed to be one of the most popular single soil surface parameters that has been related to DSC. The relative roughness index is the population standard deviation of micro and macro relives elevations for a soil plot and firstly introduced by Allmaras et al., (1966). Monteith (1974) presented the first linear model that relates DSC to RR. However, many other researchers showed that the relationship between DSC and RR is a second-degree polynomial.

Hansen et al., (1999) developed a model based on physically measured roughness index named mean upslope depression (MUD) and use it to estimate the depression storage capacity (DSC) for 32 Danish tilled soil surfaces. But he observed that MUD seemed to overestimate small DSC and overestimate large DSC values. In a field study scanning more than 200 tilled soil in Europe Kamphorst et al., (2000) compared several roughness indices including Random Roughness (RR), tortusity, limiting elevation difference and slope and mean upslope depression (MUD) with their ability to predict maximum depression storage (DSC) calculated from digital elevation model. He concluded that RR was the best indices describing the DSC linearly with coefficient of determination R2=0.8 but there is a need for different prediction model for low DSC values (Onstad,1984; Linden et al., 1988; Hansen et al., 1999).



Borselli and Torri (2010) the first to present+ a conceptual model where the rate of change of depression storage is related to the stored volume itself and the rate of change of some soil parameter, i.e. RR or $P_{100}$ of Abbot–Firestone curve. They showed by integration that the relationship between DSC and RR is exponential. They then postulated that the effect of slope on DSC will also follow an exponential decay function and that there will be always a minimum storage volume that should be included in the equation. The final relationship of their model is shown in Table 1, which shows also the models developed by other researchers.

Most of the models presented in the literature were empirical and based on statistical relationship between DSC and relative roughness and slope resulting in several fitting parameters. The validity of those models is doubted beyond the experimental values from which the model was developed. In addition, these models have been developed and tested on surfaces with large values of soil roughness, i.e. tilled soil. However, the models' performance is questionable on soil surfaces having small roughness indices, similar to those usually encountered in the maicrocatchments of rainwater harvesting in arid and semi-arid lands. Farmers used microcatchments to enhance runoff in the catchment area and channelled runoff to cropping area to enhance crop growth. The roughness of these micro catchments is expected to decrease exponentially with time due to rainfall (Zobeck and Onstad, 1987; Potter, 1990; Bertuzzi et al., 1990). Therefore, the objective of this study is to develop and test a model that integrates roughness coefficients and land slope and its ability to describe DSC under relatively small roughness values and wide range of surface slops.

## 2 Experimental methods

### 2.1 Fabrication and elevation measurement of experimental plots

Impermeable surfaces with different roughness values were constructed using cement, gypsum and four sizes of gravels, 5, 10, 20, 30 mm named P1, P2, P3 and P4, respectively. Impermeable surfaces were used to separate the influence of depression storage on runoff from infiltration (Kamphorst et al., 2000). Four wooden trays with dimension 300 x 300 x 30 mm (depth) were used and filled with a layer of cement to 15 mm depth and the gravel was fixed on the cement morter and left for 24 hours in room temperature for drying. After drying, a layer of gypsum and wax was sprayed on the cement gravel surface to create a rough and impermeable surfaces as shown in Fig. 2. Detailed is described by Abd Elbasit et al. (2009).

A 30 steel pin micro-relief device was used to measure the relative elevation for 900 points in each plot with cell size equal to 10x10 mm as described by (Abd Elbasit et al., 2009). The variation in surface elevation was marked as the variation of the pins red mark on a graph paper. Abd Elbasit et al. (2009) present detailed description for the surface construction and pin-micro-relief device. The random roughness index (RR) for each plot was calculated from the 900 elevation data as follows (Allmaras et al., 1966):



$$RR = \left[ \frac{1}{k} \sum_{i=1}^{k} (Z_i - \overline{Z})^2 \right]^{1/2} \tag{1}$$

where RR is random roughness; $Z_i$ is elevation at i point, $\overline{Z}$ is average elevation; k is number of points, in this case equal to 900. The resulting RR values for R1, R2, R3 and R4 were 0.88, 1.83, 3.91, and 6.33 mm respectively (Fig. 2)

**2.2 Measurement of Depression Storage Capacity (DSC)**

The DSC was determined under the four roughness levels and nine slopes using water mass balance between inflow and outflow water volumes (Antoni et al., 2010). In each experiment an inflow water with known value was applied at the upstream end of the plots using a disked shaped distributer device with a peristaltic water pump as shown in Fig. 3. The disk distributor dispenses inflow water to the plots evenly through 30 outlets consisting of plastic tubes and needles. At the downstream end of the plots we monitor the outflow water using graduated cylinder. The difference between steady state inflow and outflow

represents the depression the detention storage (Antoni et al., 2010). At the end of each experiments the mass of the water remained on the plot was measured and recorded as the depression storage capacity (DSC) for that surface. The experiments were carried out using nine slopes, 1, 2, 3, 4, 5, 8, 10, 15, and 20 degrees, replicated three times. The mean and standard deviation of DSC were calculated for each surface-slope combination.

**2.3 Conceptual model**

We attempted to hypothesize the shape of the relationship between DSC and RR and slope rather than developing the equation based on step wise regression. The overland flow can be described using Manning's equation for wide channel as:

$$v = \frac{1}{n} D^{2/3} S^{1/2} \tag{2}$$

$$q = v.D = \frac{1}{n} D^{5/3} S^{1/2} \tag{3}$$

where $v$ = Flow velocity, m/s; q is the unite width discharge m²/s; $D$= water depth or storage, m; $S$= Channel slope (bed slope)

and $n$ is the Manning's roughness coefficient, s/m$^{(1/3)}$. The overland flow can also be related to surface roughness $\eta$ and water depth or storage $D$ to the power $m$ that is (Stammers and Ayers, 1957)

$$q = \eta.D^m \tag{4}$$

Equating equations 2 and 3 will reveal the following proportional equation

$$\frac{D^{5/3}}{D^m} \alpha \frac{n.\eta}{S^{1/2}} \tag{5}$$

We postulated that the left hand side represents water storage with dimension L, and $n.\eta$ represents relative roughness RR to the power equal to that of the slope. We may envisage the shape of the equation by arbitrary selecting $m=2/3$ and using a





constant $\lambda$ and adding a constant $\beta$ representing the minimum storage capacity as suggested by Borselli and Torri (2010). The *DSC* equation will be as follows:

$$DSC = \lambda \frac{RR^{1/2}}{S^{1/2}} + \beta \qquad (6)$$

### 3 Results and Discussion

5   The combined depression storage (DSC) and detention storage (DS) were calculated from the difference between the inflow, outflow and compared to the residual DSC as measured at the end of each experiments. Figure 4 shows a linear relationship between DSC and combined DSC plus DS, $R^2 = 0.9$ and $P \ll 0.05$, confirming the concept presented by Antoni et al. (2010). DSC values seemed to be 24% lower than that of depression and detention. The DSC values were used in the following analysis here after.

The interrelationships between DSC, RR and slope are explored in Fig. 4, 5 and 6. First, the relationship of DSC with slope for each RR value is shown in Fig. 5. As expected the DSC increased with RR and follow a decay power function in relation to the slope with coefficient of determination R2 varied from 0.84 to 0.94 (P<0.001). The power constant ranges from -0.49 for RR=0.88 mm to 0.67 for RR of 6.3 mm. This in agreement with the investigations carried out by Burselli and Torri (2010)

who showed a decaying power relationship between DSC and plot slope using experimental data and inclined cup model. Other researchers have reported an inverse relationship between DSC and slope but these relationships were linear (Onstad, 1984; Mwendera and Feyen, 1992) or squared (Hansen et al., 1999).

Data analysis showed that DSC is negatively proportional to slope and directly proportional to RR. In fact, most of literature

researches have also confirmed this type of relationships in plot and field scale levels (Huang and Bradfrod, 1990; Onstad, 1984; Kamphorst et al., 2000; Hansen, 1999; Borselli and Torri, 2012). Although, some of these relationships and models were not properly developed resulting in negative DSC values in some case for small RR values (Onstad, 1984; Mwendera and Feyen, 1992). Therefore, we proposed a simple correlation between DSC and the ratio of RR and Slope (RR/S, mm) and presented the data on Fig. 6. As shown in the figure there is a strong and significant relationship (P<0.001 and R2=0.84)

between DSC and RR/S represented by the following power function:

$$DSC = 0.013 \left( \frac{RR}{S} \right)^{0.532} \qquad (7)$$

The previous model is not linear and has low coefficient of determination that would be expected when DSC is related to either RR or S separately. Therefore, we attempted to simplify and linearize the model by using rationally developed relationship



that relates the DSC to the square root of RR/S presented in Equation 6. The proposed relationship fits the data significantly
with R2 = 0.89 and P<0.001 as shown in Fig. 7:

$$DSC = 0.0166\left(\frac{RR}{S}\right)^{0.5} - 0.0079 \tag{8}$$

Regression test revealed that the constant -0.0079 is not significantly different than zero (p<0.01) and therefore is assumed to
be zero. Physically, the constant represents the amount of detention storage of the surface that has been removed from the DSC
measured data and therefore should be equal to zero. Therefore, we performed a second regression equation assuming the
constant equal to zero resulting in the following equation with R2=0.899, P<0.001. (Fig. 7):

$$DSC = 0.0157\left(\frac{RR}{S}\right)^{0.5} \tag{9}$$

Equation 9 is the simplest equation found in the literature and could predicts the DSC of micro catchment with low roughness
values at reasonable accuracy. Other DSC models found in the literature were mostly dealt with tilled surfaces with relatively
large roughness values. To test the validity of some of the popular models found in the literature, the DSC values predicted by
models found in the literature were plotted against the DSC predicted by the current model and the results are shown in Fig.
8. Models tends to largely over predict DSC compare to current model especially at lower DSC values (Hansen, 1999; Borselli
and Torri, 2012). Two of the models fail completely to predict low DSC values and gave negative values as shown in Fig. 8
(Onstad, 1984; Mwendera and Feyen, 1992). We postulated that these large differences between the current model and the
previous models is attributed mostly to large surface roughness the land slope. Therefore, the predicted DSC using various
models were plotted against the land slope as shown in Fig. 9 for a specific value of RR = 0.7 mm. The current model presented
in Equations 7 and 8 were quit similar and gave lower DSC values compared to other models. Hansan (1999) model seemed
to slightly under predict DSC values for RR=7 mm but failed for smaller RR values and gave negative DSC values. The effect
of slope in the current model is large when s varied from 1 to 5% but the slope effect for large S values were insignificant.
These results are in disagreement with the model presented by Hansan (1999) that shows a large linear decrease in the DSC
value with slope or with Borsello and Tori (2012) model that shows a gradual decrease of DSC values for wide range of slope
values. We believe that DSC values of low roughness surfaces will have strong dependence on small slope values compared
to high slope at which DSC variation is minimal.

**3 Conclusion**

A conceptual and a simple model has been developed to predict the surface depression storage from the square root of relative
roughness and land slope with only single fitting parameter. The model is particularly applicable on surfaces with low relative
roughness similar to those usually encountered in rainwater harvesting microcatchments. The model has been successfully
tested using impervious plots with wide range of relative roughness values and installed at various slopes with coefficient of





determination = 0.9 at probability level of 99%. The model improves our understanding to factors affecting runoff from natural surfaces and can be used as a subcomponent in future rainfall runoff models in small areas.

**Author contribution:** Abdel Baset designed the experiments and carried them out with the assistant of Ojha, Yasuda and others. Abu-Zreig developed the model concept and prepared the figures and tables. Abu-Zreig and Abdel Baset prepared the manuscript with contributions from all co-authors. In depth and continuous discussion has been carried out among all authors during the preparation of the paper and the development of the conceptual model.

**Competing interests:** The authors declare that they have no conflict of interest.

**Acknowledgement:** This research was financially supported by the Japan Society for Promotion of Science and the Global Center of Excellence for Dry Land Studies. Also we thank Mr. Shunichi Yamamoto for his help in laboratory experiments.: Impermeable surface; micro-flow simulator; surface roughness; arid regions.

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





**Table 1: Empirical relationships for depression storage capacity estimation based on soil random roughness index and slope developed in previous studies.**

| ID | Relationship | Reference |
|---|---|---|
| Model [1] | DSC = 0.112RR + 0.031RR$^2$-0.012RR.S | Onstad (1984) |
| Model [2] | DSC= 0.294RR + 0.036RR$^2$- 0.01RR.S | Mwendera & Feyen (1992) |
| Model [3] | DSC= 0.369RR - 3.76RR.S + 11.1RR.S$^2$ | Hansen et al. (1999) |
| Model [4] | DSC=0.234 RR + 0.01 RR$^2$ + 0.012 RR S | Kamphorst et al., (2000) |
| Model [5] | $DSC = 0.159 + 0.55e^{1.0011RR}e^{-0.155S}$ | Borselli & Torri, (2010) |

DSC is depression storage capacity (cm); RR is random roughness (cm) defined by Allmaras et al. (1966); S is surface slope

5   in percent.

20



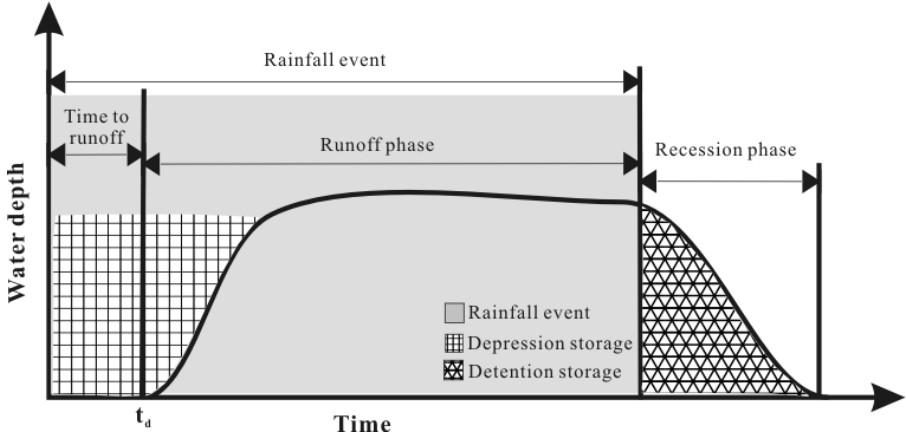

Figure 1. Conceptual Partitioning of Surface Storage into Depression Storage (DSC) and Detention Storage.

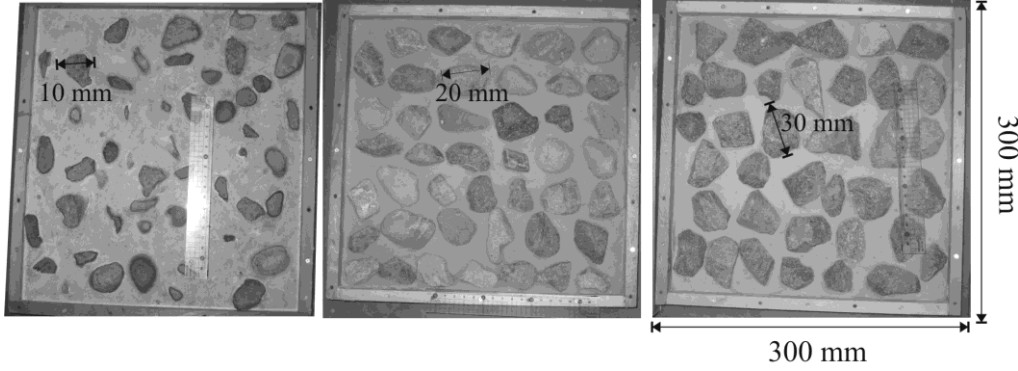

Figure 2: Fabricated Gypsum Surfaces with Three Roughness Values



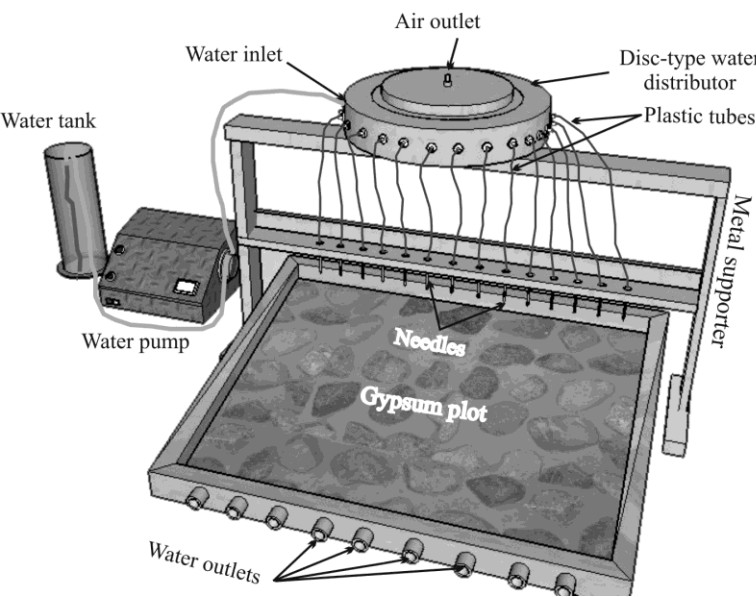

**Figure 3: Schematic View of Depression Storage in Experimental Setup Including the Peristaltic Pump, Disc-Type Water Distributor, Hypodermic Tubes, and Gypsum Plot.**





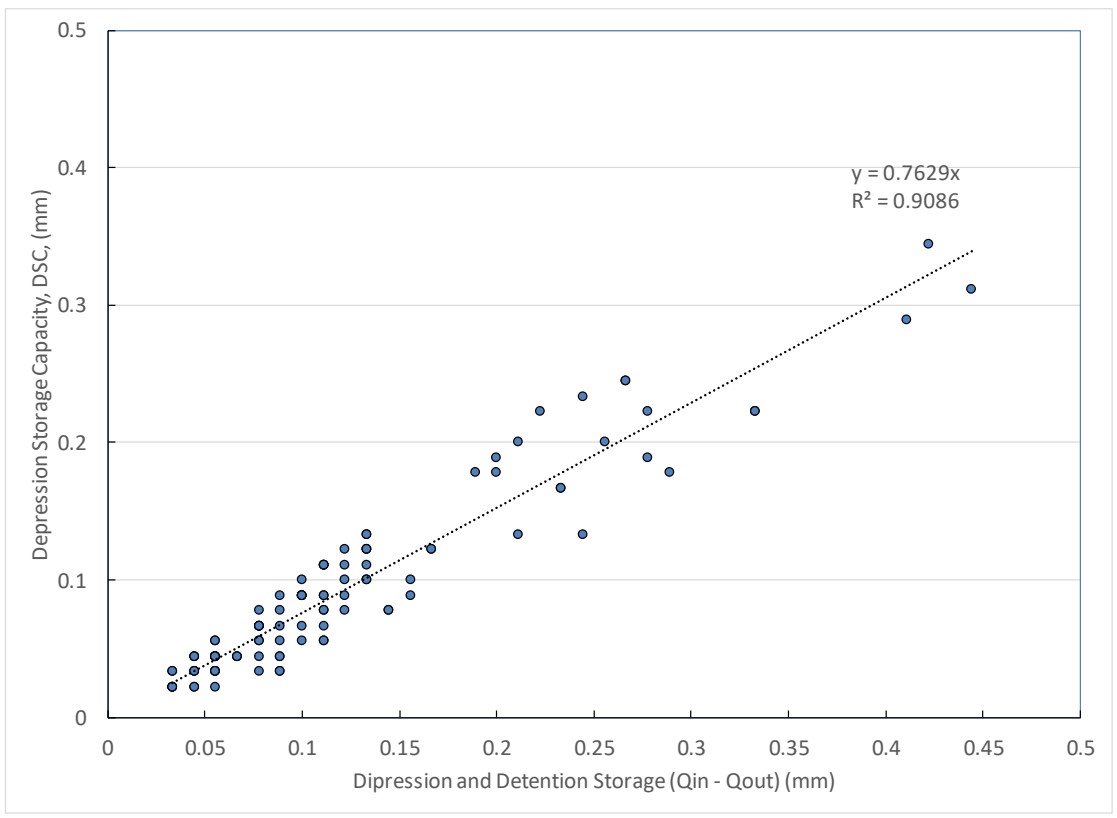

**Figure 4: Relationship between Depression Storage Capacity and Combined Depression and Detentions Storage as Measured from the Experimental Plots.**



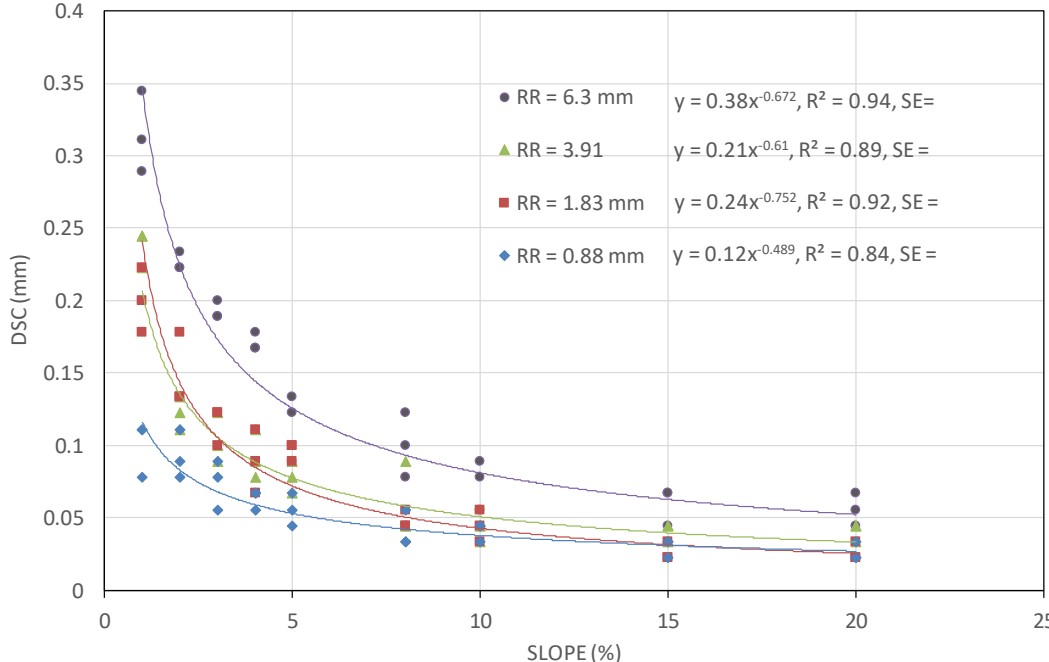

**Figure 5: Relationship Between Depression Storage (DSC) and Surface Slope with Different Surface Roughness Ranged from 0.9 to 6.3 mm.**



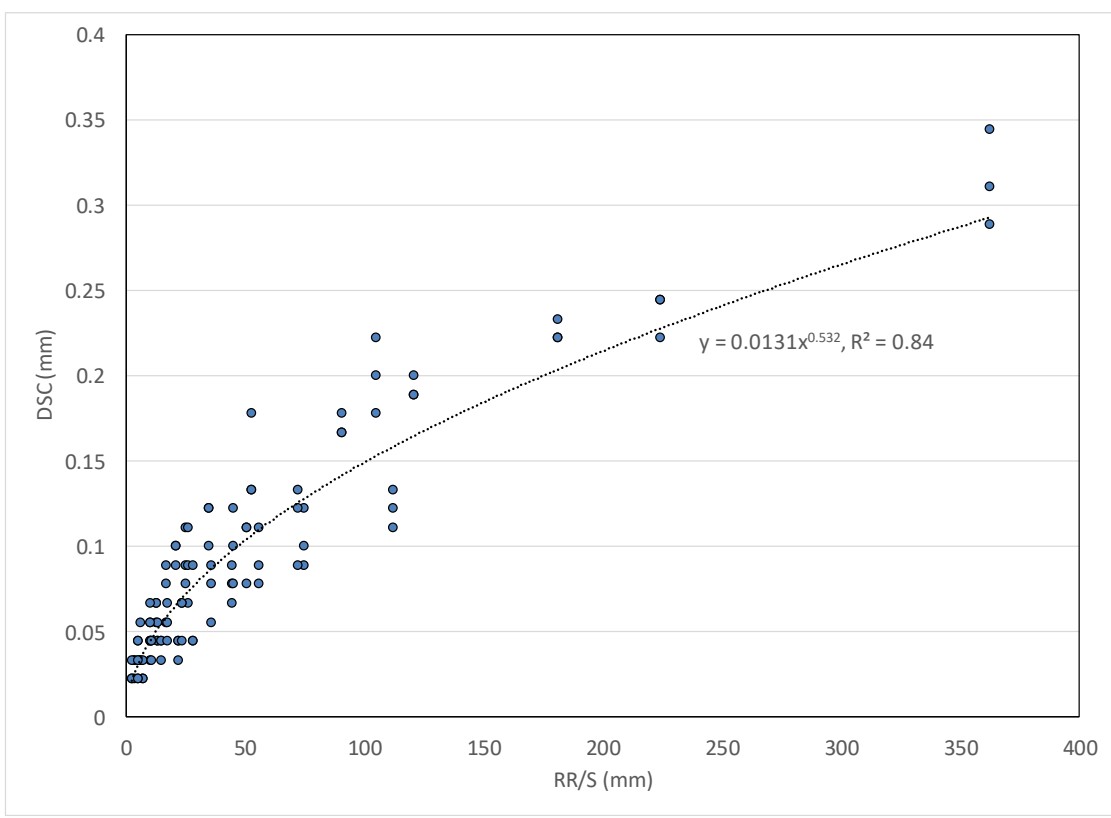

**Figure 6: Relationship Between Depression Storage Capacity and the Ratio of Relative Roughness and Land Slope (RR/S)**





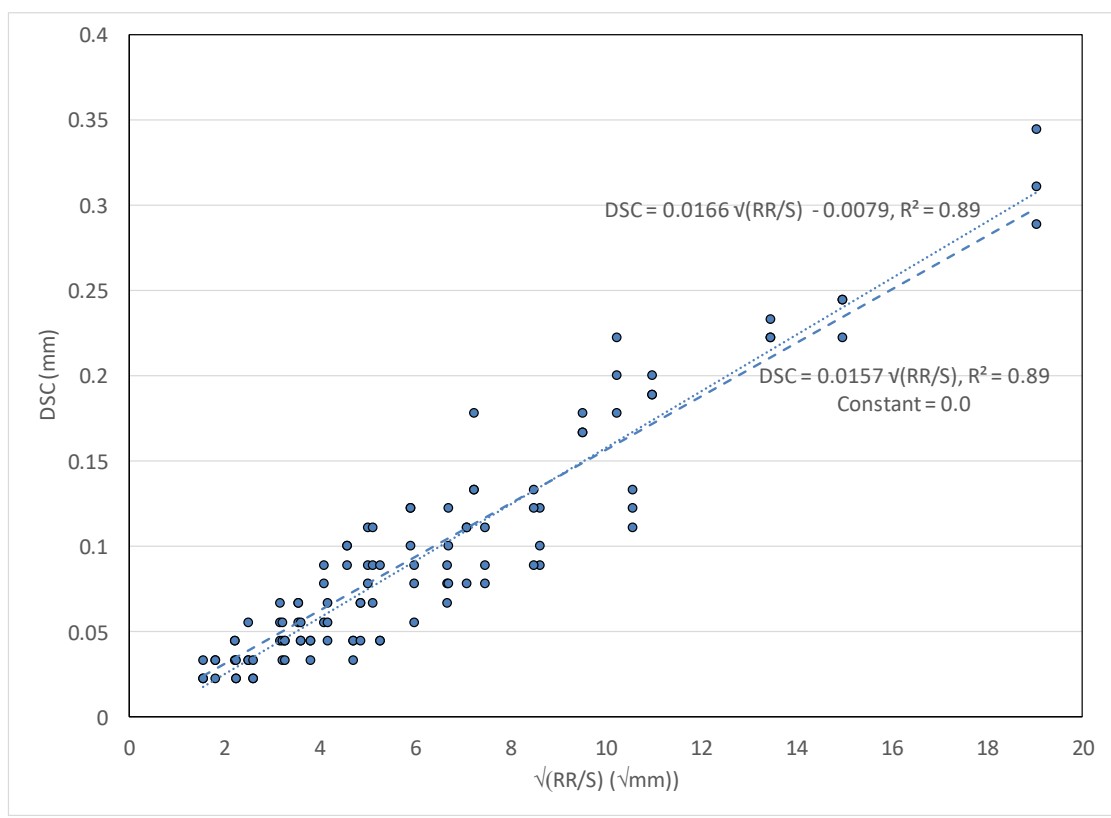

**Figure 7: Relationship Between Depression Storage Capacity and the Square Root of the Ratio of Relative Roughness and Land Slope (RR/S)[1/2]**





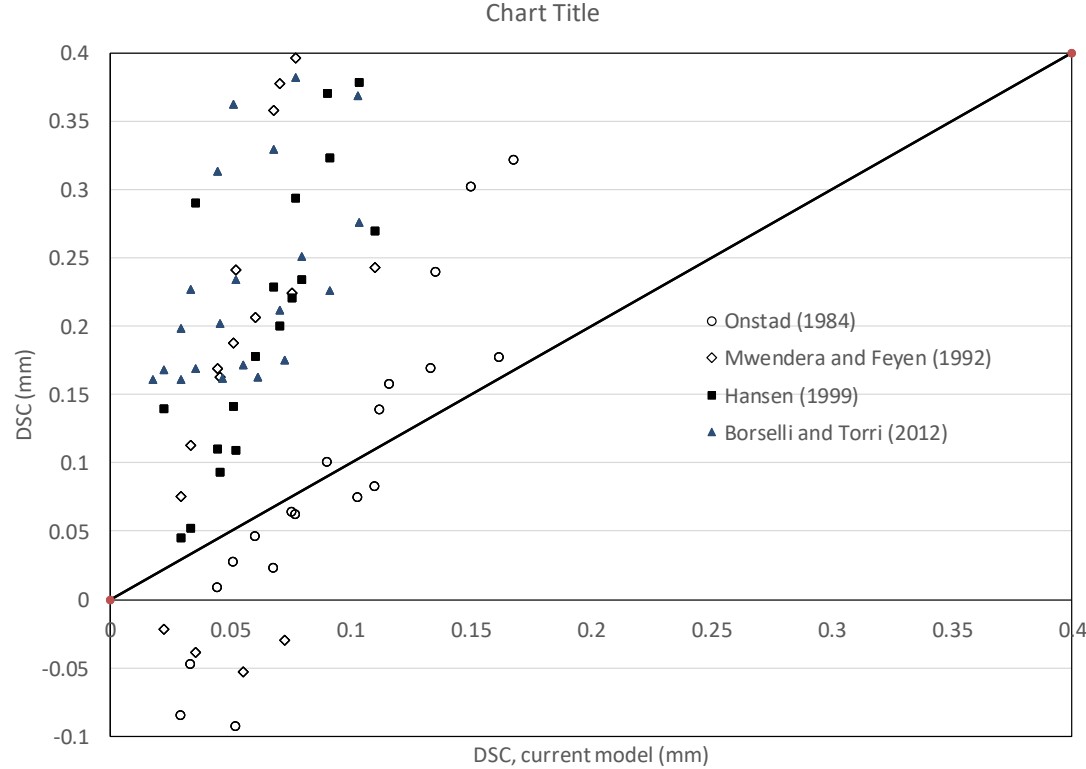

**Figure 8: Relationship Between DSC Values Predicted by Various Models Versus DSC Values of the Current Model.**




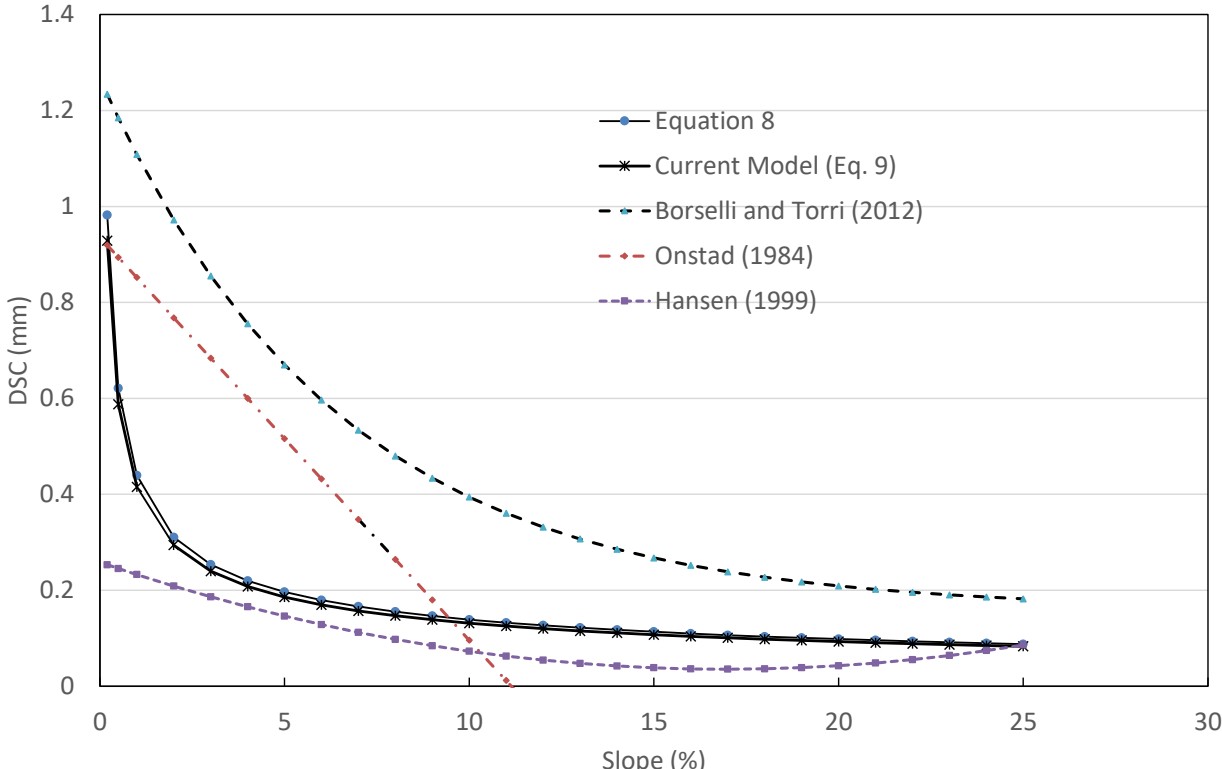

**Figure 9: The Influence of Land Slope on DSC Prediction by Various Models**