# Peer review of "Estimation of surface depression storage capacity from surface roughness"

_Hydrology and Earth System Sciences, 2019_

## Referee Comment (RC1) · Anonymous Referee #1 · 26 Mar 2019

The manuscript designed a laboratory experiment to estimate surface depression storage. However, soil surface was replaced by cement, gypsum and gravels.This treatment is quite different from the actual surface. On the agrucultural land, surface depression is mainly depended on tillage-induced microrelief, but random roughness has less effect. Therefore, for me, this model is not practical significance.

---

## Author Comment (AC1) · 30 Mar 2019

Cement and gypsum were used to seal the original soil surface so that depression storage can be separated from infiltration. This methodology has been used by several other researchers.

---

## Referee Comment (RC2) · Anonymous Referee #2 · 13 May 2019

The topic treated in this work is potentially interesting, however at my eyes too site specific to be accepted in an international journal such as HESS. Also, probably a journal closed to "Soil" should be more appropriated.

In addition, there are some major problems that can mine the entire work at its basis:

(1) the literature review seems too poor, I suggest to enlarge it looking at the recent advances in this field;

(2) a comparison with other methods available in the literature for the calculation of the surface roughness seems missed;

(3) the case study is very site-specific; it is very difficult to feel about the reproducibility of the method presented in a different context, and also its suitability to be adopted for

the analysis of the surface roughness in the field, where the local micro-morphology is more complex.

Unfortunately, because of the above issues, I suggest a rejection.

---

## Author Comment (AC2) · 18 May 2019

General Comment: The topic treated in this work is potentially interesting, however at my eyes too site specific to be accepted in an international journal such as HESS. Also, probably a journal closed to "Soil" should be more appropriated. /Response: We reported experimental results related to depression storage (DS) estimation based on a general soil roughness parameter called "Random Roughness" (RR). RR can be estimated independently of site, soil or surface condition. Therefore, it is not site specific. The current research proposed and tested a new and simple DS model that can be used to estimate DS capacity for land surfaces with relatively low RR, i.e. Fallow land compared to tilled land. The current model can be used as a subroutine in other integrated hydrological software that estimates DS. So, we think the manuscript is highly

relevant to the international scientific community.

(1) the literature review seems too poor, I suggest to enlarge it looking at the recent advances in this field; /Response: We have reported recent works (i.e. 2012) that dealt with depression storage in relation to relative roughness. We have made thorough review for the recent literature and updated the introduction section accordingly.

(2) a comparison with other methods available in the literature for the calculation of the surface roughness seems missed; /Response: The model developed here was compared with the most popular DS five models (Table 1) and the results of this comparison is shown in Figures 8 & 9

(3) the case study is very site-specific; it is very difficult to feel about the reproducibility of the method presented in a different context, and also its suitability to be adopted for the analysis of the surface roughness in the field, where the local micro-morphology is more complex.

/Response: Depression storage estimation was based on a general soil roughness parameter called "Random Roughness" (RR) that can be independently estimated for any site, soil or complex surface condition. Therefore, it is not site specific. We used gypsum-covered surfaces that were randomly arranged to create surfaces with wide range of RR values in order to separate depression storage from infiltration. Other researchers developed their depression storage models adopting this type of surfaces as explained in the manuscript.

───────────────

---

## Author Comment (AC3) · 18 May 2019

I would like to add that we indicated in the manuscript that the model is not applicable to surfaces with large RR values such as tilled soil rather for surfaces with low RR values such as fallow land for example.